



# HESS Opinions: Drought impacts as failed prospects

**Germano. G. Ribeiro Neto[1], Sarra Kchouk[2], Lieke. A. Melsen[1], Louise Cavalcante[3], David. W. Walker[2], Art Dewulf[3], Alexandre C. Costa[4], Eduardo S. P. R. Martins[5,6], and Pieter R. van Oel[2]**

[1]Hydrology and Quantitative Water Management Group,Wageningen University, Netherlands

[2]Water Resources Management Group, Wageningen University, Netherlands

[3]Public Administration and Policy Group, Wageningen University, Netherlands

[4]University of International Integration of the Afro-Brazilian Lusophony, Brazil

[5]Research Institute of Meteorology and Water Resources (FUNCEME), Brazil

[6]Federal University of Ceará (UFC), Brazil

*Correspondence to*: Germano Ribeiro Neto (germano.gondimribeironeto@wur.nl)

**Abstract.** Human actions induce and modify droughts. Yet, there remain scientific gaps regarding how anthropogenic dynamics and hydrological processes are intrinsically entangled in drought evolution. This poses the challenge of developing ways to evaluate human behavior and its pattern of co-evolution with the hydrological cycle, mainly related to water use and landscape modifications. We propose that prospect theory explains the emergence of drought impacts, such as crop losses and water shortage, if they are considered as failed welfare expectations ("prospects") due to water shortage. This behavioral economic theory is dominantly applied to explain decision-making processes under uncertainty. We argue that it can also contribute to explaining socio-hydrological phenomena such as reservoir effects. This new approach can contribute to bridging natural and social sciences perspectives for more integrated drought management that takes into account the local context.

## 1 Introduction

During fieldwork conducted by the authors of this paper in the Semiarid region of Brazil (SAB), a farmer was asked how the historic 2012-2018 multi-year drought event (Marengo, 2020; Cunha et al., 2019a, b, 2018) affected his livelihood and welfare. The farmer responded by asking: "Drought? What drought?". We wondered how a drought event that lasted for almost 7 years and was characterized by an average 60% reduction in annual precipitation had gone unnoticed by someone who had been in the middle of it. A spatial contextualization helped us answer this question. The farmer's property was located at the edge of an upstream reservoir with low water abstraction that retained water throughout this drought event, therefore, he never experienced water insecurity during this period.

The farmer's simple response can illustrate the concept of "Drought in the Anthropocene" (Van Loon et al., 2016b), which underlines the need to consider the human component as an inseparable part of the complex and interrelated processes of a drought. This concept requires more balance between the analysis of the physical and human component of drought events. In this context, drought is defined as an exceptional period of lack of water compared to normal conditions, which allows us to consider acute water shortages caused by human actions as a drought event.

These ideas are developed in the context of socio-hydrology, which proposes to change the conventional methodological framework applied to studies of disasters related to hydrometeorological extremes (e.g. droughts, floods and their variations). This field aims to study the dynamics and co-evolution of human-water coupled systems, with one of the main premises that human actions are an endogenous part of the hydrological cycle (Sivapalan et al., 2012, 2014; Pande and Sivapalan, 2017). In other words, people interact with the hydrological





system in various ways (e.g. water consumption and landscape modification) and this has the potential to alter
hydrological processes, which in turn influence and impact human actions, creating a co-evolution.
There is already solid evidence that human actions can modify, intensify and induce drought events (Van Loon et
al., 2022; Ribeiro Neto et al., 2022, 2021; Savelli et al., 2022, 2021; AghaKouchak et al., 2021). Thus, it is apparent
that the natural sciences do not provide all the necessary means to analyze droughts and the right way forward is
interdisciplinarity, especially the integration with social sciences (Di Baldassarre et al., 2018; Massuel et al., 2018;
Martin-Ortega, 2023). The reconsideration of the human component opens the opportunity to study this kind of
disaster from the bottom up, taking as a starting point the impacts that individuals in the hydrological system
experience/cause and the decisions they make to avoid these impacts (Walker et al., 2022).
Correspondingly, it is possible to expand the definition of an impact and how it is intertwined in the emergence
and propagation of a disaster considering the perspectives of individuals. The combined dynamics of these
individual behaviors result in macro-scale consequences that generate changes in the system, making it possible
to analyze the emergence of patterns not observed at other dimensions, such as hydrological variables, and spatio-
temporal scales (Wens et al., 2021, 2019; Van Oel et al., 2012). These patterns can be referred to as Socio-
hydrological Phenomena, they arise in different places around the world, in different contexts, and are often
portrayed as counter-intuitive or paradoxical (Di Baldassarre et al., 2019). Yet every phenomenon or process can
be considered as such when one does not have all the necessary tools to analyze them.

## 59     2 Impacts as failed prospects

Satisfying our needs for welfare and not just survival is one of the characteristics that define us as humans. It is
crucial to improve our understanding of how this influences decision-making related to water use and landscape
modification to better assess drought. It is important to understand that human beings as individuals anticipate the
desirable level of welfare then choose among the possible prospects they believe have the highest chance of
achieving this goal (Kahneman and Tversky, 1979). These prospects are the decision options that are associated
with an expected outcome within a scenario of uncertainties.
The kind of prospects chosen defines how well an individual is adapted to the environmental conditions in which
they are inserted, being directly related to their vulnerability and resilience. We propose that when an individual
has a failed prospect due to a lack of water situation, which can be influenced by hazards (mainly hydroclimatic
trends) and/or human actions, negatively affecting the level of welfare, then they will feel the impact and
consequently perceive the situation as a drought. For example, a prospect can be the choice a farmer makes to
grow a certain crop rather than another, to achieve greater gains or fewer losses depending on the context. This
choice is made with the expectation that this crop will contribute to the achievement of the aimed welfare level.
If, for instance, the prospect is to grow a water-consuming crop in a region characterized by low water availability,
it can be an indication of maladaptation and vulnerability of the individual. In this example, if a precipitation deficit
occurs (hazard) and this negatively affects the chosen crops resulting in an unsatisfactory production (failed
prospect), the individual will feel the impact and consider this event to be a drought. It is up to society to decide
when a set of individuals impacted by a water shortage is sufficiently serious for such an event to be considered a
drought.
Returning to the real example of the farmer mentioned above. He never had any failed prospect during that drought
event, mainly because he had a secure water source throughout this period and consequently his level of welfare
was not affected. Considering this, the simple answer he gave us is coherent and logical, since he did not experience
impacts related to the drought event that occurred in that region and therefore for him this drought event never
existed. This is yet another example that demonstrates the limitation of evaluating drought events by only
considering methods that do not incorporate impacts and ignore the human component (Kchouk et al., 2021).


## 3 Socio-hydrology and prospect theory

Considering drought as the collective impacts that emerge as failed prospects due to lack of water make it necessary to predict how individuals choose which prospects are more attractive to follow. Prospect theory emerges (Kahneman and Tversky, 1979) (PT) as a descriptive technique that explains how individuals choose alternatives when the outcome is uncertain (Tversky and Kahneman, 1986). This theory has been widely debated, especially in the socio-economic sciences, and in the environmental sciences has been applied to different fields such as reservoir operation (Bahrami et al., 2022), disaster management (Osberghaus, 2017), and irrigation water resources management (Wang et al., 2022).

One of the new concepts that PT presented is that individuals in the real world do not maximize total wealth, but react to possible or perceived gains or losses, which are emotional and short-term. In other words, human beings do not necessarily seek to maximize their net benefits, or utilities, by always choosing the prospects that produce the highest level of benefits (Jones, 1999). To clarify this concept, we invite the reader to participate in a simple experiment (Kahneman and Tversky, 1979) consisting of choosing one of the options in the following two problems: 1) 80% chance of winning $4000 or 100% chance of winning $3000; 2) 80% chance of losing $4000 or 100% chance of losing $3000.

If you chose the second and first options in problems 1 and 2 respectively, you behaved like most people who participated in such an experiment (Kahneman and Tversky, 1979). This means that you presented "risk aversion" behavior when the prospects are related to certain gains (problem 1) and "risk seeking" behavior when the prospects are related to certain losses (problem 2). The combination of these two patterns illustrates the idea presented by PT that is the human tendency to overvalue a certain outcome (or highly likely), relative to outcomes that are probable (Kahneman and Tversky, 1979; Edwards, 1996; Levy, 1992). Problem two indirectly illustrates another concept presented by the PT which is the Loss aversion effect. This one highlights the asymmetry in an individual's perception of gains and losses; thus, losses feel more "painful" than gains of equal magnitude feel "pleasurable". The consequences can be the preference for the status quo and the acceptance of riskier prospects to avoid certain losses ("risk seeking" behavior).

The definition of whether the outcome of a prospect is seen as a gain, or a loss is assessed by comparing the prospect with a Reference point which can be influenced by what is experienced as the status quo or the 'normal' situation, but also by the way the decision problem is portrayed (Kahneman and Tversky, 1984). This is called the framing effect whereby depending on how individuals perceive and make sense of decision prospects in terms of gains or losses, they will show a tendency towards risk aversion or risk seeking respectively.

Here we argue that the onset and propagation of human drought impacts, which we consider to be those that negatively affect the individual's welfare, and some socio-hydrological phenomena, can also be explained through the lens of prospect theory. The relevance of the concept of human drought impacts as a failed prospects becomes more evident when the emergence and propagation of the impacts are placed at the center of drought assessment studies. In this sense, it can be considered that this disaster arises from the moment a hazard (natural or human-related) results an anomalous lack of water that generates negative impacts, which can be social-economic (human), or environmental, and ceases when these damages are no longer observed.

The first idea to consider from PT is the Reference point concept, which is the general term for the starting point for making different kinds of decisions. For drought assessment, we consider the Reference point as the minimum welfare level that individuals tolerate to feel satisfied and secure with the results of chosen prospects, and deviations from this are what define a gain or loss. The individual's perception of their environment defines the Reference point which guides the expectations regarding their level of welfare and therefore for choosing the prospects to achieve them. This perception is influenced by environmental conditions (e.g. water availability), previous experiences (e.g. past drought events), community interactions, socio-economic trends (e.g. production costs, goods prices, local culture and governance), and it can change over time. The higher the Reference point,





the greater the potential for human drought impacts once it is not achieved. Fig. 1 presents an overview of how
prospect theory is related to socio-hydrology phenomena and drought emergence.
Once the individual has defined their Reference point and delineates the desired level of welfare, they evaluate the
decision prospects for achieving it. When faced with a situation of high-water availability, individuals have more
freedom to choose prospects that offer certain gains (risk aversion behavior, blue cycle Fig. 1) even if this promotes
a reckless water use pattern and/or the development of activities that are not necessarily the most adapted to the
environmental conditions of the region where they are inserted. Successive gains associated with this behavior, in
the short term, will reinforce the selected prospect and, in the long term, raise the Reference point. Levels of
welfare below the Reference point will be perceived as losses and avoided, even though the individual may have
already experienced such levels as a gain in a previous situation (Framing effect).
A series of successful prospects keep the upward trend in the Reference point, and this is maintained as long as
the water resources to which the individual has access can sustain their water demand. This continues even if there
is an impending drought situation, since the reduction in water consumption while the Reference point is associated
with satisfactory water availability can be framed by individuals as a direct decrease in welfare, hence a certain
loss which is avoided. When water is lacking and it is no longer possible to maintain the water-consumption
standards that the individual required, this results in failed prospects and, consequently, drought impacts arise.
Initially, the drought situation is typically perceived as a loss, as we consider that it starts after a failed prospect.
In the short term, individuals tend to focus on prospects that can at least prevent further losses, even if they were
previously seen as risky (risk seeking behavior, orange cycle Fig. 1). However, in the long term, if the low water
availability persists, it can cause individuals to adjust their expectations by lowering the Reference point. In other
words, individuals can be less impacted by water shortages simply because they accept suboptimal outcomes (e.g.
lower agricultural production or productivity). Once this shift in Reference point occurs, individuals may no longer
view the situation as a drought, but rather as the "new normal".

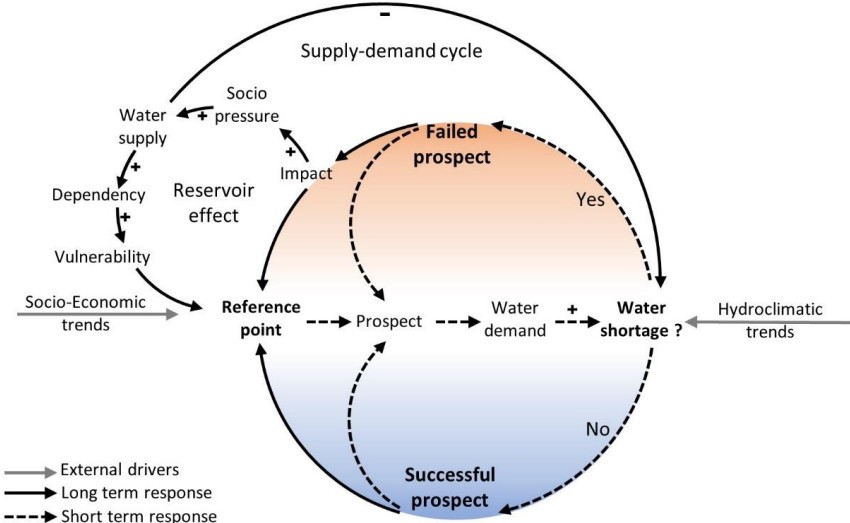


**Figure 1. – The cycle of human drought impacts.** Our hypothesis emphasizes the centrality of the human component (starting
from the Reference point) in the emergence of drought impacts with the individual as the primary scale. Moreover, the
combination of how they link to the hydroclimatic and socio-economic trends results in the emergence of long-term socio-
hydrological dynamics (reservoir effects and supply-demand cycle) that can be explained by concepts related to Prospect theory
such as: Reference point; Framing effect; Risk aversion (blue cycle) and risk seeking (orange cycle) behavior.



As water availability gradually increases, either due to natural causes (hydroclimatic trends) or due to the
expansion of water infrastructure, individuals are likely to shift away from their lower Reference point and search
for prospect that offer more certain which restarts a new cycle (blue cycle Fig. 1). We hypothesize that the demand
to expand the water infrastructure can be related to when individuals attribute the occurrence of drought impacts
to low water availability without considering the suitability of their own chosen prospects in local environmental
conditions. This behavior can then, in the long term, result in social pressure to increase water supply (e.g. reservoir
construction and water transfer), and when this is met, individuals can re-enter the cycle of increasing water
consumption (blue cycle, Fig,1). As the demand continues to rise, it can eventually offset the new maximum supply
capacity. This can lead to more social pressure to increase water availability, thereby creating a vicious cycle
(Supply-demand cycle, Fig.1), greater dependence on water infrastructure, and greater vulnerability to drought
events (Reservoir effect, Di Baldassarre et al., 2018, Fig.1).
**4 Prospect theory and drought - insights from the Brazilian semiarid region**
The 2012-2018 drought event in the Semi-Arid region of Brazil (SAB) is used as a practical example that highlights
how prospect theory fits into the narrative of this kind of disaster. Here we will focus on Ceará state, which was
one of the sub-regions most impacted by this event. Fig. 2 presents the percentage anomaly of annual precipitation
relative to the long-term climatological average (1981-2011) of SAB and Ceará state (magenta polygon) during
the 2012-2018 drought event. The years prior to this drought were characterized by precipitation levels above the
climatological average, which meant that most reservoirs in Ceará had stored volumes close to their maximum
capacity.
This region has a historical susceptibility to drought events and in recent times, there has been observable change
in the preparation and management of such disaster. This change was related to a shift from a "fighting against
drought" perspective, which relied on hard solutions such as significant investments in water infrastructure, to
"cope with drought" perceptive which relies on soft solutions such as renewed focus on public policy towards
adaptative measures and integrated water resources management (Cavalcante et al., 2022; Medeiros and Sivapalan,
2020). Nevertheless, the high water availability experienced during the previous years to the 2012-2018 drought
contributed to the support of high water demand productive activities, such as rice paddies and irrigated fruit crops.

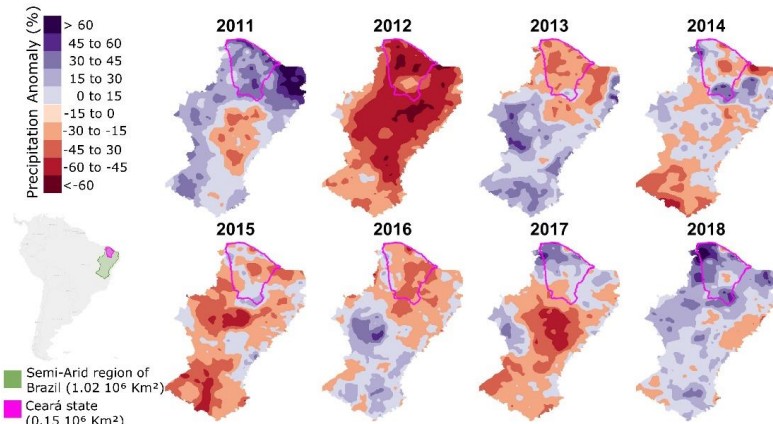

**Figure 2. Precipitation variability in the Semi-Arid of Brazil during the drought 2012-2018**. Percentage anomaly of annual
precipitation relative to the long-term average (1981 to 2011) using the Climate Hazards center InfraRed Precipitation with
Stations (CHIRPS, Funk et al., 2015) dataset available on https://www.chc.ucsb.edu/data. The location of Ceará state (magenta
outline) and the Semi-Arid region of Brazil (green outline) are presented in relation to South America as well as their respective
areas.
Before the occurrence of this drought, Ceará had already been experiencing a gradual growth of dairy cattle
farming which was intensified during this event. Farmers increasingly started to see this activity as a prospect more
adapted, from a local perspective, to droughts because it guarantees a source of perennial income and serves as a
capital reserve (part of the herd can be sold at any time). Furthermore, it is considered that cattle farming is less
dependent on locally produced inputs and on the spatio-temporal heterogeneity of the precipitation regime when
compared to rainfed crops.
We hypothesized, based on field interviews, that periods of high water availability provided a certain stability to
farmers who depended on rainfed crops. However, the following and more frequent occurrence of intense drought
events caused them to experience consecutive production losses (failed prospects) which led them to view dairy
production as a prospect that would avoid further losses. One of the barriers that made individuals view this activity
as unattractive or risky was the low and volatile price of a liter of milk in the local market. This changed when
associations of small dairy producers were created, and they started to have more bargaining power within the
dairy industry. In this new socio-economic configuration, individuals begin to see cattle farming as a prospect
more adapted to drought and which promotes more certain gains. This is further evidenced by farmers who had
already adopted this activity due to previous drought events and that continued to favor this kind of prospect even
in later periods of greater water availability.
The expansion of dairy production in Ceará has resulted in the increase of small (informal) reservoirs to support
forage production and provide water for livestock consumption. In some regions the high concentration of such
structure can reduce the hydrologic connectivity of the watershed, impacting the recharge of large reservoirs
downstream that serve multiple purposes and prolonging the hydrological drought impacts (Ribeiro Neto et al.,
2022). As result, the persistence of a low water availability condition can influence the individuals' perception of
the environment and, consequently, their definition of the Reference point.
Interviews with farmers and agricultural extension officers regarding desirable reservoir volumes illustrated the
concept of the Reference point and how it can vary according to previous experiences. Reservoir levels as low as
20% of capacity were unexpectedly celebrated. Interviews revealed that volumes were consistently around 5%
during the 2012-2018 drought; the lower water availability had become the status quo (or the Reference point).
Therefore, increased volumes up to 20% of capacity were considered gains even though such a level would have
been considered a loss prior to the multi-year drought.
Based on the case study presented here we can exemplify some situations that can be analyzed under the Loss
aversion effect concept. We consider that such patterns are related to the attempts of individuals to adapt to drought,
aiming in general to avoid greater losses through measures that reduce water demand. We observed that one of
these adaptations was the search for hybrid bovine breeds resulting from the crossing of local breeds that are
resistant to drought with European breeds that have a higher milk production. These hybrid breeds were already
known by the local farmers, but they were seen as a not worth investment, due to the high cost of acquisition.
However, during the last drought, an acceleration in herd replacement by these hybrid breeds was observed. Many
farmers decided to sell part of their herd to raise capital to invest in these hybrid breeds. They realized that it would
be safer, in a scenario of low water availability, to maintain a smaller but more productive herd.
The increase in the number of wells in Ceará during this drought event is another practical example that illustrates
the Loss aversion effect concept. For Ceará, this alternative water supply can be considered a risky prospect, as it
presents high implementation costs and is associated with uncertainties to whether a viable water resource will be
found for exploitation. Either because of the water quality (brackish groundwater is common) or because
crystalline geology often provides low yield. Therefore, it is perceived that individuals in this region who chose to
install wells were willing to take more risks to avoid greater losses.



## 5 Simulating prospect theory effects - applications, challenges and opportunities

The failure to consider patterns of co-evolution between hydrological processes and human dynamics within a hydrological system was rooted mainly in the fact that human dynamics were considered insignificant to cause noticeable consequences and due to the low spatio-temporal resolution at which hydrological models originally operated. Implicitly there was the idea that it would be impossible or unfeasible to implement anthropogenic actions as an intrinsic component of the hydrological cycle, which has been successively refuted by various studies related to drought assessment (Wens et al., 2021, 2019; Van Oel et al., 2012; Streefkerk et al., 2023; Wens et al., 2020; Bakarji et al., 2017; Van Oel et al., 2018).

The concept presented here of (human) drought impacts as failed prospects provides a different perspective to incorporate into the analyses of the socio-hydrological characteristics of each region. This can contribute especially to the improvement and development of drought monitoring and early warning systems, socio-hydrological characterization, drought risk analysis, forecast/re-analysis of drought events, and support the development of public policies for the mitigation and prevention of drought impacts. On the other hand, the prospect theory has limitations mainly related to the failure to explain how decision-making, especially related to the definition of an individual's Reference point, is influenced by the environment and the full range of affective and emotional states.

As argued above, we consider that when applied to drought assessment, the Reference point is related to the minimum level of individuals' well-being to feel satisfied with the outcome of the chosen perspectives. To represent this concept, it is necessary to study the evolution of human dynamics mainly related to how water and land have been used over time by individuals in the hydrological system. Agent-based models (ABM) are a promising framework for this kind of study, as they allow explicit probabilistic simulation of human decision-making with the ability to respond, learn and adapt to variations in environmental states and other agents (Schrieks et al., 2021). Moreover, it has been successfully applied in socio-hydrological studies mainly combined with hydrological and/or agricultural models (Wens et al., 2021, 2019; Streefkerk et al., 2023). These types of analyses often require expertise usually associated with the social sciences, such as interviews, workshops, companion modelling, and serious games (Massuel et al., 2018; Acosta-Michlik and Espaldon, 2008; Pouladi et al., 2019; van Duinen et al., 2016). This further underline that drought assessment studies are conceptually interdisciplinary and therefore require solutions beyond those associated only with the natural sciences.

The possibility of explaining the occurrence of a drought event through the concepts of Prospect theory, which was initially presented to explain human behavior in economic decision-making, endorses the importance of the human component in drought assessment, besides bringing new discussions on this topic. The core concept presented here advocates for a greater focus on the human component within drought assessment studies and places the emergence of human impacts as a precursor to the disaster. The reconsideration of what drought impacts are and how they occur through the concepts of prospect theory allows us to consider that drought is first and foremost a socio-hydrological phenomenon that materializes in the form of a disaster.

There is already an understanding and acceptance of the concept of "human-induced", "climate-induced" and "human-modified" droughts (Van Loon et al., 2016a) that explore the main causes that trigger different types of drought events. Drought impacts as failed prospects concept does not refute these terminologies, since they are useful in indicating the main forces that are disrupting the hydrological system and causing the anomalous water shortage that characterizes a drought event. Nor does it invalidate established concepts of definition or classification of this disaster such as 'agricultural drought' and 'hydrological drought', as these terminologies relate to the main types of impacts that individuals suffered during the analyzed event.

The hypothesis presented here can contribute to the identification of new socio-hydrological phenomena and improve the understanding of others already described in the literature. Furthermore, it contributes to the call for a change of perspective on how studies related to disasters of hydro-meteorological extremes, especially drought events, should be conducted, bringing new ideas about the importance of incorporating the human component in



these issues. Finally, we also support the idea of bringing more balance between the "socio" and "hydro"
component in the studies related to drought assessment, in which more interdisciplinarity should be sought as
hydrology and meteorology alone simply do not provide the means to understand human dynamics within the
(socio-)hydrological cycle.
**Competing interests**
The contact author has declared that none of the authors has any competing interests.

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
