# Peer review of "HESS Opinions: Drought impacts as failed prospects"

_Hydrology and Earth System Sciences, 2023_

## Author Response (AR1)

**RC1 – Sivapalan**

I read this paper with considerable interest.

Firstly, I am a co-author of a paper that looked at asymmetric drought response from the viewpoint of prospect theory (Tian et al., 2019), which the authors may want to cite as supportive evidence.

Secondly, I have a direct interest in the coevolutionary change (humans and water) in the Ceara region of Brazil (Medeiros and Sivapalan, 2020, which is cited in the paper), and am presently trying to also study and model this behavior (publications forthcoming). I am encouraged that the authors think that prospect theory might be a suitable explanation for what has been observed in the Ceara.

For these reasons, I am generally supportive of eventual publication of this paper in HESS. But I am also disappointed that the paper is presented as an opinion paper. If they already think of prospect theory as a plausible hypothesis, why not do the analysis and test it as a hypothesis? Why do they present it as an opinion paper?

Given that it is presented as an opinion paper, and the authors successfully argue that it can only be an opinion paper, I go to the next step of reviewing it as an opinion paper.

In the paper, after an introduction, they first present prospect theory in the form of a tutorial, and then present some patterns from Ceara to highlight the observed phenomena. For an opinion paper this type of organization is somewhat lacks punch.

My suggested organization is to present the introduction focused on droughts and drought propagation. I will then present the history of what happened in the Ceara, in a way that makes it clear what the "phenomenon" is. If at all possible, the phenomenon should be presented using a signature that automatically or implicitly makes one suspect that prospect theory might apply. This can then be followed by a description of prospect theory for those people that may not be aware of it. Finally, the paper can conclude with what you plan to do about it – or what anyone else should do about it, so the paper is shown to serve a stated purpose.

The problem I have with the current structure is that it comes across as "solution looking for a problem": prospect theory comes first, Ceara (data evidence) comes afterward. The scientific method works from analysis of data (data on drought propagation in the Ceara), then extracting of some pattern (e.g., asymmetric water consumption), which leads to a hypothesis (prospect theory).

I hope this helps. Even if the authors do not follow my advice, I hope they can make the presentation a bit more insightful and interesting to read.

References Cited

Tian, Fuqiang, You Lou, Hongchang Hu, W. Kinzelbach and M. Sivapalan (2019). Dynamics and driving mechanisms of asymmetric human water consumption during alternating wet and dry periods. *Hydrological Sciences Journal,* Vol. 64(5), pp. 507-524. doi: 10.1080/02626667.2019.1588972

Medeiros, P. H. A. and M. Sivapalan (2020). From hard path to soft path solutions: Slow-fast dynamics of human adaptation to droughts in a water-scarce environment. *Hydrological Sciences Journal*, Vol. 65(11), pp. 1803-18014

*We thank the reviewer for dedicating time to conduct this review and for endorsing the scientific relevance of our study. The reviewer raised three main points in the review (reason for choosing the opinion paper format, structure of the manuscript, and readability), which we address below.*

*1 - The choice for an opinion paper*

*We believe this manuscript fits the opinion-paper format of HESS, even though it is not a classical opinion but mainly a perspective. The HESS description of Opinion papers is: "They are discussed openly in HESSD so as to stimulate an open debate among peers on new ideas, views, or perceptions in hydrology". We propose a new view of looking at drought impacts: from the bottom up, using prospect theory (supported with empirical evidence from the field). As such, we believe that this falls under the umbrella of an HESS opinion paper.*

*2 -The structure of the manuscript*

*We thank the reviewer for the suggestion on restructuring the manuscript. We have evaluated the suggestion but decided we would prefer not to follow the suggested structure. The main reason is that it would result in more repetition, thereby increasing the length of the manuscript; first explaining the phenomenon, then introducing prospect theory and then again linking it to the phenomenon. We do, however, acknowledge the feeling of the reviewer that it reads as a theory looking for a problem. We also acknowledge the lack of "punch". Through rephrasing and restructuring parts of the text we hope to take away this feeling and add this punch, indeed by including hints going forward. In this regard, we suggest the following modifications:*

*"Although the patterns of co-evolution between the human component and the hydrological cycle have been widely debated in the scientific literature (Sivapalan et al., 2012; Di Baldassarre et al., 2015; Van Loon et al., 2016b; Di Baldassarre et al., 2019; Tian et al., 2019), gaps remain regarding the relationship between hydrological hazards (e.g., drought), the perception of impact of this hazard, and occurrence of the hazard itself. With the ideas presented in this paper we aim to contribute to this discussion, focusing on drought hazards.*

*We argue that the collectivity of individuals' perception of the impacts that they experience determines the magnitude and the very occurrence of a drought event, this being related to both environmental and socio-economic factors. Using Prospect theory (Kahneman and Tversky, 1979), stemming from the field of behavioral economics, we can explain the emergence of drought impacts, considering impacts as failures in expected welfare due to water shortages. We build our case by first presenting the concept of drought impacts as failed prospects, then the relationship between socio-hydrology and Prospect theory to finally present how this can be applied to real cases of drought events." (Lines 52 to 64 revised manuscript)*

*3 – Readability*

*This feedback is in line with many of the suggestions and questions from reviewer 2. We have done a thorough language and text editorial round to increase the readability and clarity of the manuscript.*

**RC2 – Anne Van Loon**

This is a very interesting opinion paper. It focuses on water security and thereby addresses an important and timely topic. The authors make a case for viewing the co-evolution of people and the water cycle through the lens of prospect theory. I think it is very useful to explore how social science theories can be used in socio-hydrology to explain phenomena related to drought and water shortage. I see potential for this paper to be published.

The core idea of this paper is interesting and the conceptual figure and example are nicely presented. I really like the statements in relation risk-seeking and risk-aversion and how these decisions are related to their (perception of the) environment and their previous actions.

But some justifications and explanations should be clarified. For example, it is unclear to me what, in the theory presented, here is the connection between drought perception, human influence on drought hazard, and drought impacts on people. This could be made more explicit. I also got a bit confused about the Reference point and how the authors see this in relation to drought (they define it as welfare, but later talk about the environment as Reference point). And I also have a few questions about what in the authors' view the relation is between the individual and society. I think this could be improved by a more distinct phrasing and more examples of what the authors mean.

They can also make a stronger case about the supply-demand cycle and the reservoir effect being explained by the proposed concept of applying prospect theory to drought.

Below I give examples and some suggestions of how to improve the clarity and message of the paper.

*We thank the reviewer for dedicating time to conduct this review and for endorsing the scientific relevance of our study. We also thank the reviewer for bringing these relevant issues to our attention, which are answered one by one.*

**Specific comments**

The relation between hazard and impact should be made more clear, for example in the abstract. It now starts with a discussion on the influence of human activities on drought hazard, but then moves on to talk only about drought impacts. This connection is more clear in the Introduction section of the paper (p.2, l.48-50: "The reconsideration of the human component opens the opportunity to study this kind of disaster from the bottom up, taking as a starting point the impacts that individuals in the hydrological system experience/cause and the decisions they make to avoid these impacts"). But it does not convince me why including the human component opens up the opportunity to study droughts in a bottom up way.

*We thank the reviewer for the comment and agree that the phrasing of the sentences referred to was not clear enough. In this regard, we suggest the following modifications for the abstract and to the paragraph mentioned, respectively:*

*"Yet, there remain scientific gaps regarding how hydrological processes, anthropogenic dynamics and individuals' perception of impacts are intrinsically entangled in drought occurrence and evolution." (Lines 13 to 15 revised manuscript)*

*"Perceiving the human component as an inseparable part of the hydrological cycle creates new research avenues, for instance to study drought events and other disasters at scales that are commonly disregarded. For example, by starting from the individuals in the hydrological system that experience impacts, and by evaluating the decisions they make to avoid these impacts. This may reveal the emergence of patterns and phenomena unobserved at other spatio-temporal scales or when focusing on other hydrological variables (Wens et al., 2021, 2019; Van Oel et al., 2012; Walker et al., 2022)." (Lines 47 to 52 revised manuscript)*

On page 2, lines 81-82 there is another explanation ("he did not experience impacts related to the drought event that occurred in that region and therefore for him this drought event never existed"). This is about perception of hazard based on impact, not on human modification of the hazard yet (although the reservoir is anthropogenic, it is not clear that the farmer made the reservoir and therefore that there is a link between the human influence on hazard and impacts).

*We thank the reviewer for the comment and propose the following modification to improve the clarity of the text:*

*"The farmer's response implicitly reveals the relationships between human actions that modify hydrological processes (in this case, the construction of a reservoir) which alter exposure to a drought hazard (in this case, no exposure because of a filled reservoir) and how this in turn influences individuals' own perceptions of disaster occurrence ("Drought? What drought?"). This is in line with the concept of "Drought in the Anthropocene" (Van Loon et al., 2016b), which underlines the need to consider the human component as an inseparable part of the complex and interrelated processes of a drought." (Lines 33 to 38 revised manuscript)*

P.3 l.116-122: The reasoning for why drought impacts can be explained with prospect theory (and not with other theories) is not yet completely clear to me. Please be a bit more specific / give more examples.

*From the interviews in the field it became clear that there is a mismatch between the common indices used to identify drought events and the impacts experienced on the ground. Based on that, we advocate for the idea that the starting point of drought analysis should be the impact occurrence at the individual level. Prospect theory is a useful framework to explain the dynamics at this individual level, while also being able to explain emergent behavior at a higher level, such as the reservoir effect. Other theories might be valuable as well, in fact, the prospect theory lens on drought might uncover a wealth of other potentially useful theories. We have reread the text and emphasized this perspective at several places throughout the text.*

P.3 l.124-127: Here the Reference point is considered as "the minimum welfare level that individuals tolerate to feel satisfied and secure with the results of chosen prospects". But if so, how is the minimum welfare level related to the "individual's perception of their environment". It feels (again) like the authors do not explain the relationship between hazard or perception of hazard (or environment) and impact (or welfare).

*We thank the reviewer for bringing this relevant concern to our attention and propose the following modification to clarify this issue:*

*"For drought assessment, we consider the Reference point as the minimum welfare level that individuals tolerate to feel satisfied and secure with the results of chosen prospects, and deviations from this are defined as a gain or loss. The environment guides the individuals' expectations regarding their level of welfare (Reference point), and with that for choosing the prospects to achieve them. For instance, the Reference point can be influenced by environmental conditions such as water availability, which is related to aspects of food and water security, previous experiences (e.g. past drought events), community interactions (e.g. peer comparison), and socio-economic trends (e.g. production costs, goods prices, local culture and governance). Importantly, the Reference point will vary over space and time. For instance, a higher yield loss might be incorporated as acceptable in the Reference point after years of drought, or in a region with consequent insecure water supply." (Lines 126 to 134 revised manuscript)*

I have the same issue in the description of the Brazil case p.6 l. 212-213, which states after a description of how farmers increasingly do dairy production and build small reservoirs for livestock, that "As result, the persistence of a low water availability condition can influence the individuals' perception of the environment and, consequently, their definition of the Reference point." Here, I miss the link from the perception of the environment back to the Reference point (minimum welfare level accepted). And in l.217 the water availability is the Reference point.

*We have applied the following modification:*

*"As a result, the persistence of this hydrological impact affects the region's water availability, which in turn can influence individuals' perception of water security (component of welfare) and consequently their definition of the Reference point." (Lines 219 to 222 revised manuscript)*

P.7 l.266-269: "places the emergence of human impacts as a precursor to the disaster. The reconsideration of what drought impacts are and how they occur through the concepts of prospect theory allows us to consider that drought is first and foremost a socio-hydrological phenomenon that materializes in the form of a disaster." What is a disaster if not drought impacts? How are the authors reconsidering what drought impacts are? And did we not already see drought as a socio-hydrological phenomenon without considering prospect theory?

*We agree that the wording of the quoted paragraph is not clear enough and therefore have applied the following change:*

*"The core concept presented here advocates for a greater emphasis on the human component in drought assessment studies, positing the emergence of human impacts, rather than solely hydro-meteorological ones as a precursor to such a disaster. This viewpoint contrasts with the methodological approach of numerous studies in which drought events are analyzed only considering the spatio-temporal variability of hydrometeorological variables, disassociated from the human component. Furthermore, the Reference point concept provides a theoretical basis for considering drought impacts dynamically, in contrast to the static vision of drought impacts that is now often encountered in drought assessment studies. Prolonged drought impacts lead to a change in the individuals' perception of drought occurrence, the impacts have become the new "normal" situation and are therefore no longer experienced as impacts. Moreover, we argue that the concept of drought impacts as failed prospects reinforces the perspective that drought is first and foremost a socio-hydrological phenomenon that materializes in the form of a disaster." (Lines 272 to 282 revised manuscript)*

To address these unclarities, I would suggest the authors to carefully check their phrasing and structure to guide the reader a bit more in how the prospect theory can be applied to drought.

What I think would also help with the clarity of the paper is if the authors would expand Figure 1 to three panels, one with the original prospect theory, one with the application to drought impacts and adaptation (the figure they have now), and one with the Brazil example.

*Based on the questions and remarks above and the feedback from Reviewer 1 we have thoroughly gone through the manuscript to improve the text, both in terms of structure and in terms of formulation. We will evaluate the structure (a suggestion of Reviewer 1) in line with the figure suggestion here of Reviewer 2. Related to this we propose the addition of Figure 3, which would be inserted in Section 4. This figure shows the application of prospect theory concepts to the Brazil case study. We have applied the following modification to link the text to the new figure.*

*"Fig. 3 presents an overview of Prospect theory applied to the Ceará study case. We hypothesized, based on field interviews, that periods of high water availability provided a certain stability to farmers who depended on rainfed crops (short term positive response, first blue dashed arrow, Fig 3). However, the following and more frequent occurrence of intense meteorological drought events caused them to experience consecutive production losses (failed prospects) which led the individuals to view the exclusive production of rainfed crops as a riskier prospect (short term negative response, red dashed arrow, Fig. 3) and dairy production as a prospect that would avoid further losses (long term negative response, red arrow, Fig. 3). One of the barriers that made individuals view this activity as unattractive or risky was the low and volatile price of a liter of milk in the local market. This changed when associations of small dairy producers were created, and they started to have more bargaining power within the dairy industry. In this new socio-economic trend, individuals began to see cattle farming as a prospect more adapted to drought and which promotes more certain gains (short term positive response, second blue dashed arrow, Fig. 3). This is further evidenced by farmers who had already adopted this activity due to previous drought events and that continued to favor this kind of prospect even in later periods of greater water availability (long term positive response, second blue arrow, Fig. 3)." (Lines 200 to 213 revised manuscript)*

[Figure]

**Figure 3. Prospect theory in socio-hydrology applied to Ceará study case.**

**Textual comments:**

- 2. l.51-52: I don't understand this sentence. What was your definition of impact and how are you suggesting to change it?

  *We thank the reviewer for this comment and agree that the definition of an impact should be clear and objective. We believe we did this later in the text (115-116) however to avoid having to present this definition already in the introduction which would also require the definition of other concepts, we decided to delete the sentences referred to in this comment (51-52).*

- 2 l.76-78: I follow the reasoning for using prospect theory at the individual level, but I cannot follow the step from individual to society. Is there a certain threshold of individuals being impacted that affects the welfare of a society such that they consider an event a drought?

  *We meant that even though a drought arises at the scale of the individual, it is still somewhat validated by society (whether through regulatory/research agencies, or government). We have applied the following reformulation:*

  *"if a precipitation deficit occurs (hazard) and this negatively affects the chosen crops resulting in an unsatisfactory production (failed prospect), the individual will feel the impact and consider this event to be a drought. If there is a critical mass that experiences impacts, this might lead to the (official) declaration of a drought. This is the result of a complex interaction involving many factors: those experiencing impact, their societal position, media exposure, power-relations, the political consequences of formally declaring a drought, et cetera." (Lines 82 to 85 revised manuscript)*

  3 l.117: "and some socio-hydrological phenomena" > which socio-hydrological phenomena do you mean? Explain or leave out.

  *We have applied the following modification:*

*"We argue that the onset and propagation of human drought impacts (which we consider to be those that negatively affect the individual's welfare), and socio-hydrological phenomena (e.g. the reservoir effect and supply demand cycle), can be explained through the lens of prospect theory." (Lines 121 to 123 revised manuscript)*

- 5 l.184: "during the previous years to the 2012-2018 drought" > "during the years previous to the 2012-2018 drought"

    *Accepted and modified.*

- 6 l.211: "prolonging the hydrological drought impacts" or "prolonging the hydrological drought" (hazard)?

    *We understand that the high concentration of reservoirs referred to in the sentence in question can prolong the hydrological impacts of drought as well as the occurrence of the event itself, however for ease of reading we will adopt "prolonging the hydrological drought".*

    Thanks for your responses. I think these are all good suggestions and I'm curious how you are planning to modify the figure.

**RC3 – Anne Van Loon**

Just one small issue that I noticed: you mention the "hydrological impacts of drought", but it is not completely clear what you mean here. The societal impacts of hydrological drought or the effects of meteorological drought on water resources (i.e. hydrological drought).Looking forward to seeing the revised version of your manuscript.

*We thank the reviewer for the comment. When we mentioned "hydrological impacts of a drought" we were referring to the impacts commonly associated with the so-called hydrological droughts. We have applied the following modifications to improve the clarity of the text:*

*"In some regions the high concentration of small reservoirs decreased the hydrologic connectivity of the watershed, impacting the recharge of large reservoirs downstream that serve multiple purposes (Ribeiro Neto et al., 2022). As a result, the persistence of this hydrological impact affects the region's water availability, which in turn can influence individuals' perception of water security (component of welfare) and consequently their definition of the Reference point." (Lines 217 to 222 revised manuscript)*

---

## Author Response (AR2)

Rebuttal
HESS-2023-136

HESS Opinions: Drought impacts as failed prospects

Both Referees agree that the paper was improved through revision, but provide constructive comments that should be addressed. In addition to the technical comments of Referee #2, I agree with the point made by Referee #1. Indeed, the paper would be enriched by adding a concluding section turning "the technical language of prospect theory into an opinion/conclusion in simple language that an average reader (hydrologist) can easily understand". Moreover, it important to stress that prospect theory is not a solution looking for a problem, but the other way around. Prospect theory as an elegant hypothesis for explaining for what has been observed in Brazil. The abstract should also be revised accordingly.

*We would like to thank the editor and reviewers for the review process and for recognizing the relevance of our study. We present below the changes made in response to the reviewers' suggestions.*

[revised manuscript text omitted]